# Diet Supplemented with Special Formula Milk Powder Promotes the Growth of the Brain in Rats

**DOI:** 10.3390/nu16142188

**Published:** 2024-07-09

**Authors:** Ruiqi Mu, Jufang Li, Yu Fu, Qinggang Xie, Weiwei Ma

**Affiliations:** 1Capital Medical University, School of Public Health, Beijing Key Laboratory of Environmental Toxicology, Beijing 100069, China; muruiqi1999@163.com (R.M.); fuyu000319@163.com (Y.F.); 2Feihe Reseach Institute, Heilongjiang Feihe Dairy Co., Ltd., Beijing 100015, China; lijufang@feihe.com (J.L.); xieqinggang@feihe.com (Q.X.)

**Keywords:** infant milk powder, growth and development, brain, lipid metabolomics

## Abstract

This investigation was to study the effects of different formula components on the brain growth of rats. Fifty male SD rats were randomly divided into five groups: a basic diet group; a 20% ordinary milk powder group; a 20% special milk powder group; a 30% ordinary milk powder group; and a 30% special milk powder group by weight. LC-MS was used to detect brain lipidomics. After 28 days of feeding, compared with the basic diet group, the brain/body weights of rats in the 30% ordinary milk powder group were increased. The serum levels of 5-HIAA in the 30% ordinary milk powder group were lower than in the 20% ordinary milk powder group. Compared with the basic diet group, the expressions of DLCL, MePC, PI, and GM1 were higher in the groups with added special milk powder, while the expressions of LPE, LdMePE, SM, and MGTG were higher in the groups with added ordinary milk powder. The expression of MBP was significantly higher in the 20% ordinary group. This study found that different formula components of infant milk powder could affect brain growth in SD rats. The addition of special formula infant milk powder may have beneficial effects on rat brains by regulating brain lipid expression.

## 1. Introduction

It is recognized that breast milk is the best source of nutrition for infants [1]. Breast milk is the first and main source of nutrition for infants and is rich in nutrients, vitamins, and biologically active compounds that contribute to the development and growth of infants [2], including the development of the immune system and the gut microbiota. Not only that, but the health benefits extend to breastfeeding mothers, as breastfeeding can prevent ovarian and breast cancer [3]. Given the strong impact of breast milk on a child’s and mother’s health, the World Health Organization (WHO) advised that babies should receive only breast milk for the initial six months of their existence. Apart from the nutritional advantages, breastfeeding has many other advantages, such as being convenient and inexpensive and, more importantly, a bonding experience for both a mother and a baby. The choice to breastfeed is very individual and is frequently impacted by numerous factors. In certain circumstances, breastfeeding may not be feasible, inappropriate, or insufficient, which may necessitate stopping or discontinuing breastfeeding [4]. Despite the recommendations of the WHO, only a minority of infants worldwide, less than half, are being exclusively breastfed for the first six months of their lives. As a result, they are relying on infant formula for their nutritional needs from a very young age [5].

Infant formula is commonly used to replace breast milk for infant feeding [6]. Milk is typically utilized as the foundational element of the formula, with additional components incorporated to closely mimic the makeup of human breast milk and garner nutritional advantages [4,7], such as fatty acid structure, amino acid composition, and the type and composition of oligosaccharides. These elements support proper growth and maturation of the intestines, microbiota, as well as the enhancement of brain function and the immune system.

The initial two years of life are acknowledged as crucial for developing cognitive skills and behaviors that endure throughout one’s life [8], in which the human brain is sensitive to environmental stimuli and develops at a high speed [9]. Nutrition plays a crucial role in this process, with research indicating that differences in dietary intake during this critical period can impact brain structure and function, leading to potential implications for mental health conditions [10]. Therefore, it is important to have formulas with ingredients that are suitable for infant brain development.

This experiment was conducted to compare the biological indexes of weaned rats fed with different formulas and different doses of milk powder. In order to find out whether the special formula satisfies the basic nutritional needs of infants and young children and its effect on brain growth, different doses were set to be investigated. This study will provide clues for subsequent epidemiological studies and provide a basis for promoting the brain growth of infants and young children.

## 2. Materials and Methods

### 2.1. Animals

Fifty male SD rats of the SPF class were weaned at 3 weeks of age (21–27 days), corresponding to 6 months of age in humans. The intervention was conducted with different types and doses of milk powder for 4 weeks, corresponding to the postnatal to pre-pubertal period in humans. During 7 days of adaptive feeding, all the rats were fed a maintenance diet (calories from proteins: 22.47%; calories from carbohydrates: 65.42%; calories from fat: 12.11%; 3.42 kcal/g). All rats were supplied by HFK Bioscience Co., Ltd. (Beijing, China). Then, the rats were randomly divided into five groups according to their body weights: a basic diet group; a 20% ordinary milk powder group; a 20% special formula milk powder group; a 30% ordinary milk powder; and a 30% special formula milk powder group. The experimental animals were purchased from Beijing Viton Lihua Laboratory Animal Technology Co., Ltd. under license No. SYXK (Beijing, China) 2022-0049 and SCXK (Beijing, China) 2021-0011. All animal procedures were approved by the Animal Care and Ethics Committee of Capital Medicine University (Ethics Review No: AEEI-2023-035), and the approval was passed on 26 March 2024. The flow chart of the experimental design is shown in Figure 1.

### 2.2. Diet

Basic diet, 20% ordinary milk powder diet, 20% special milk powder diet, 30% ordinary milk powder diet, and 30% special milk powder diet. Special formula refers to milk powder with appropriate amount of special ingredients, including Docosahexaenoic Acid (DHA), Eicosatetraenoic acid (ARA), galactose oligosaccharide, 1,3-dioleic acid-2-palmitate triglyceride, lutein, nucleotide, lactoferrin, animal Bifidobacterium, etc., while ordinary milk powder in this experiment did not have the above special ingredients added. Table 1 shows the difference in composition between the two types of milk powder. The feeds were supplied by KOA (Tianjin, China) Feed Co., Ltd. (Guangzhou, China) Ordinary milk powder and special formula milk powder were replaced with corn starch components at two different doses of 20% and 30%. The energy ratio and energy density of the three major macronutrients are essentially the same in the five groups (calories from proteins: 17.3%; calories from carbohydrates: 63.9%; calories from fat: 18.8%; 3.99 kcal/g). Table 2 shows the dietary energy ratios and formulations of five groups of rats.

### 2.3. MRI Analysis of the Brain

After the rats were anesthetized intravenously and placed in a prone position, their heads were placed in a magnetic resonance imaging (MRI) coil, and the differences in the water signals of the left and right cerebral hemispheres in the MRI images were used to assess whether there was any damage to the cerebral structures. MRI technique was used to scan the anesthetized SD rats for nuclear magnetic resonance diffusion tensor imaging (DTI), DTI is a method to quantify the diffusion motion of water molecules in three-dimensional space by measuring their diffusion intensity in all directions, thus obtaining the values of a variety of parameters, of which the most commonly used parameters are FA (fractional anisotropy), apparent diffusion coefficient (ADC), and mean diffusion (MD).

### 2.4. Biological Sample Collection

On the 35th day of group feeding, after anesthetizing the rats by intraperitoneal injection of tribromoethanol, blood was collected from the heart. After euthanizing, the rats were sacrificed, and the brain tissues were collected rapidly. The entire brain was rinsed repeatedly with 0.9% saline, then dried on filter paper, weighed, and stored in a cryogenic freezer at −80 °C.

### 2.5. Serum Extraction and ELISA

Blood was collected by cardiac sampling using tubes free of any pyrogens and endotoxins and was allowed to clot naturally for 20 min at room temperature. The serum was separated by centrifuge at 3000× *g* for 20 min. Rat 5-hydroxytryptamine (5-HT) and 5-hydroxyindoleacetic acid (5-HIAA) were detected in the specimens by double antibody sandwich enzyme-linked immunosorbent assay (ELISA). The purified rat 5-HT and 5-HIAA capture antibodies were coated on a microtiter plate to prepare solid-phase antibodies. Rat 5-HT and 5-HIAA were sequentially added into the coated microtiter wells and combined with HRP-labeled detection antibody to form an antibody–antigen–enzyme-labeled antibody complex, which was washed completely and then washed and added with the substrate TMB to develop the color. The absorbance at 450 nm (OD value) was measured with an enzyme marker, and the standard curve was plotted to calculate the content of 5-HT and 5-HIAA in rat serum in the sample.

### 2.6. Paraffin Sides and HE Staining

Paraffin sectioning was first performed. The fresh tissue was fixed with fixed liquid for more than 24 h, then it was trimmed and put in the dehydration box. The dehydration box was put into the dehydrator in order to dehydrate with gradient alcohol. The wax-soaked tissue is embedded in the embedding machine and cooled on a −20 °C freezing table, and after the wax is solidified, the wax block is removed from the embedded frame and repaired. The trimmed wax block was cooled on a −20 °C freezing table, and the modified tissue chip wax block was sliced on the paraffin slicer; the slice thickness was 4 μm. Then, paraffin sections were dewaxed: Xylene I for 20 min–Xylene II for 20 min–100% ethanol I for 5 min–100% ethanol II for 5 min–75% ethanol for 5 min–rinsed with tap water. Sections were stained with Hematoxylin solution for 5 min and rinsed with tap water. Then, the sections were treated with a Hematoxylin Differentiation solution and rinsed with tap water. Then, the sections were treated with Hematoxylin Scott Tap Bluing and rinsed with tap water. After the sections were treated with 85% ethanol for 5 min and 95% ethanol for 5 min, the sections were stained with Eosin dye for 5 min. Finally, the tissue sections were dehydrated, sealed, and microscopically examined, and images were captured and analyzed, and Image-Pro Plus 6.0 software was applied to select five intact villi per section at a 100× scale and measure the length of the villi (μm) and the depth of the crypt fossa (μm), respectively.

### 2.7. Lipidomic Analysis of Brain Tissue

Firstly, 50 mg of the sample was accurately weighed into a 2 mL centrifuge tube. Then, a 6 mm diameter grinding bead, 280 µL of extraction solution, and 400 µL of MTBE were added into the tube. The beads were ground on a frozen tissue grinder (−10 °C, 50 Hz) for 6 min. Then, the sample was extracted for 30 min (5 °C, 40 KHz) and left at −20 °C for 30 min. Then, the sample was centrifuged (13,000× *g*, 4 °C); 350 µL of supernatant was placed into an EP tube and dried under nitrogen, and 100 µL of extraction solution (isopropanol: acetonitrile = 1:1) was added to re-dissolve. The sample was ultrasonically extracted at low temperature for 5 min (5 °C, 40 KHz) and centrifuged for 10 min (13,000× *g*, 4 °C), and the supernatant was transferred to the injection vial with an internal cannula for analysis. In addition, 20 µL of supernatant was pipetted from each sample and mixed to create quality control samples. The instrument platform for this LC-MS analysis was a UHPLC-Q Exactive HF-X system (Thermo Fisher Scientific, Beijing, China) for ultra-high performance liquid chromatography tandem Fourier transform mass spectrometry (UHPLC-Q). The chromatographic conditions were as follows: an accucore C30 column; the mobile phase A was 50% acetonitrile in water (containing 0.1% formic acid and 10 mmol/L ammonium acetate); and the mobile phase B was acetonitrile/isopropanol/water (10/88/2) (containing 0.02% formic acid and 2 mmol/L ammonium acetate). Finally, quality control (QC) samples were prepared by mixing the extracts of all the samples in equal volumes, and the volume of each QC was the same as that of the samples, which were processed and detected by the same method as that of the analyzed samples, and one QC sample was inserted into every 5–15 analyzed samples during the instrumental analysis in order to examine the stability of the whole detection process.

### 2.8. Detection of Protein Expression in Rat Brain Tissue by Western Blot

The brain tissues were lysed in Radio Immunoprecipitation Assay (RIPA) lysis buffer (Beyotime, Shanghai, China). A BCA total protein assay kit was used to determine the concentration of protein (APPLYGEN, Beijing, China). Equal amounts of protein extracts (30 µg) were separated by 12% SDS–polyacrylamide gel and transferred onto polyvinylidene difluoride membrane (Millipore, Bedford, MA, USA). The membrane was blocked with 5% nonfat milk for 1 h at room temperature and then incubated with primary antibodies for β-actin (Cell Signaling Technology, Boston, MA, USA), PSD-95 (Cell Signaling Technology, USA), PLP (Bioss, Beijing, China), MBP (Cell Signaling Technology, USA) overnight at 4 °C. Subsequently, at room temperature, the membrane was incubated with horseradish peroxidase-conjugated secondary antibodies (Jackson, Lancaster, PA, USA) for 1 h. The membrane was washed with TBST 3 times for 10 min each time. ECL (Millipore, Bedford, MA, USA) was added dropwise to the protein side of the membrane and reacted for 5 min; the film was exposed for 5 min, developed for 2 min, and then fixed. The results were quantified as the ratio of the relative gray value of the target protein to the internal control, β-actin.

### 2.9. Statistical Analysis

SPSS 21.0 software was utilized to statistically analyze the data in order to obtain relevant information. In order to analyze the experimental data, descriptive statistics were used, and quantitative indicators were expressed using mean ± standard deviation (mean ± SD). For the comparison of the five groups of measures, one-way ANOVA was used for data that conformed to normality and had chi-square variance; then, Tukey’s multiple comparisons test followed; non-parametric tests (Kruskal–Wallis test) were used for data that did not conform to normality, and Welch method and Dunnett’s T3 method were used for data that conformed to normality but did not have chi-square variance, where *p* < 0.05 indicated a statistically significant level of significance of the differences. GraphPad Prism 8.0 was used for graphing.

## 3. Results

### 3.1. Brain/Body Weight Ratio Index of SD Rats

As shown in Figure 2, After 28 days of group feeding, there was a significant difference in the brain weight/body weight of the five groups of rats (*p* = 0.011). The 30% ordinary milk powder group increased the brain weight/body weights of rats compared to the basic diet group (*p* = 0.032).

### 3.2. Serum 5-HT and 5-HIAA Content of SD Rats

As shown in Figure 3a, after 28 days of feeding, there was no significant difference in serum 5-HT content between the five groups of rats.

Compared with the 20% ordinary milk powder group, the serum content of 5-HIAA of rats in the 30% ordinary milk powder group was increased, and the difference was statistically significant (*p* = 0.005).

### 3.3. The Structure of the Brain of SD Rats

The results of He-stained sections showed that the overall lesions in the brain tissues of SD rats in the five groups were mild, and the main lesions were neuronal degeneration. As shown in Figure 4, there was no statistically significant difference in the counts of neuronal degeneration and necrosis in the brain tissues of the different groups of rats.

As shown in Figure 4a, MRI images showed that there was no obvious damage to the brain tissue of the rats in each group; the FA values of the rats in each group were between 0 and 1, and there was no obvious change, indicating that the white matter fibers of the brain were not damaged. There were no obvious differences in the levels of MD in each group, and there were no obvious differences in the values of ADC, so this study showed that the cells of the brain tissue of the rats in each group were not obviously damaged.

### 3.4. The Lipidomics in SD Brain

#### 3.4.1. Comparative Analysis of Lipid Metabolites in the Five Groups

By the classification method of LIPID MAPS, lipid metabolites were classified into eight major classes: fatty acyls (FA); glycerol esters (GL); glycerophospholipids (GP); sphingolipids (SP); sterolipids (ST); isoprenoid lipids (PR); glycolipids (SL); and polyketides (PK), which were, in turn, subdivided into 96 subclasses. As shown in Figure 5, the numbers and names of lipids identified for each subclass detected in this experiment, with GP and SP accounting for a larger proportion and ST accounting for a smaller proportion. As shown in the Venn diagram of Figure 5b, the number of lipid metabolites shared by the five groups of rat brain tissues was high, with MePC (22:6/14:3) being the lipid metabolite unique to the brain tissues of rats in the 20% of the special milk powder group. PCA (Principal Component Analysis) showed that there was a high degree of similarity of samples within the five rat groups and that the group with the addition of 20% special milk powder was more different from the other groups in terms of types of lipid metabolites, with a cumulative difference explanatory R2X (cum) = 0.772. A supervised analysis method called PLS-DA (Partial Least Squares Discrimination Analysis) led to the same conclusion, as shown in Figure 5d; the group with the addition of 20% special milk powder was more different from the other groups in terms of types of lipid metabolites, and the model’s predictive power was Q2 = 0.846.

#### 3.4.2. Analysis of Samples of Five Groups Differential Lipid Metabolites

GP is one of the most abundant lipids in the cerebral cortex. In Figure 6a, results showed the relative concentration and type changes in GP in the brains of rats in different groups. The relative concentration of dilysocardiolipin (DLCL) in the 30% special milk powder group was higher than that in the basic diet group (*p* = 0.013). The relative concentration of Methyl phosphatidylcholine (MePC) in the 20% special milk powder group was higher than in the basic diet group (*p* = 0.015). Similarly, the relative concentration of phosphatidylinositol (PI) was higher in the 20% special milk powder group than in the basic diet group (*p* = 0.013). The relative concentration of lysophosphatidylethanolamine (LPE) was lower in the 20% special milk powder group than that in the basic diet group (*p* = 0.005), and the relative concentration of lysodimethylphosphatidylethanolamine (LdMePE) was, likewise, lower in 20% special milk powder group than that in the basic diet group (*p* = 0.019).

The variations in the abundance of SP in the brains of mice in different groups are shown in Figure 6b. The relative concentration of Gangliosides (GM1) in the 30% ordinary milk powder group was higher than the basic diet group (*p* = 0.028), and in the 30% special milk powder group, the relative concentration of GM1 was higher than the basic diet group (*p* = 0.006). Compared with the basic diet group, the relative concentrations of sphingomyelin (SM) in the 30% ordinary milk powder group were significantly higher (*p* = 0.040).

The expression of triglyceride (TG) and diglycerides (DG) in GL indicated almost no difference between the five groups (*p* > 0.05), but the relative concentration of Monogalactosyldiacylglycerol (MGDG) in the basic diet group was lower than that in the 20% ordinary milk powder group (*p* = 0.031) and the 30% ordinary milk powder group (*p* = 0.046).

The intake of different formula milk powders had a great effect on the saturation of FA in the brain. As shown in Figure 7, the majority of lipids were polyunsaturated with at least one double bond. Lipids with one unsaturated were the most abundant, and fully saturated lipids were the third. The relative concentration of fully saturated lipids in the 30% special milk powder group was higher than that in the 20% special milk powder (*p* = 0.011). The relative concentration of lipids containing three double bonds was lower in the 30% special milk powder group than in the basic diet group (*p* = 0.008). The relative concentration of lipids containing six double bonds was lower in the 30% ordinary milk powder group than in the basic diet group (*p* = 0.018). The relative concentration of lipids containing seven double bonds was lower in the 20% ordinary milk powder group than in the basic diet group (*p* = 0.042).

#### 3.4.3. KEGG Pathway Analysis

Table 3 shows the top ten metabolic pathways involved in the number of lipid metabolites. As shown in the table, rat brain tissue lipid metabolites are mainly involved in the metabolic pathway, glycerophospholipid metabolism pathway, and choline metabolism pathway in cancer, as well as retrograde endogenous cannabinoid signaling, insulin resistance, fat digestion and absorption, glycerol ester metabolism, sphingolipid signaling pathway, sphingolipid metabolism, and the pathway in cancer.

### 3.5. Brain Tissue Protein Expression in SD Rats

The Western blotting results in Figure 8 showed that the differences in expression levels of PSD-95 and PLP in the brain tissues of five groups of rats were not statistically significant. Compared with the basic diet group, MBP protein expression levels in the brain tissues of SD rats were significantly increased in the 20% ordinary milk powder group (*p* = 0.0168) and significantly decreased in the 30% ordinary milk powder group (*p* = 0.0141) and the 30% special formula milk powder group (*p* = 0.0025) compared with the 20% ordinary milk powder group.

## 4. Discussion

According to the World Health Organization, undernutrition is associated with 2.7 million child deaths worldwide, which makes up 45% of all infant mortality. LMICs in Sub-Saharan Africa and South Asia bear the majority of the burden, with 99% of infant mortality cases. About 250 million children under five face the threat of inadequate development and growth stunting. Thus, providing sufficient nutrition is essential for fostering children’s growth and development [11].

It is widely acknowledged that the initial 1000 days of life, starting from conception up to 2 years after birth, represent a crucial phase in the development of infants [12,13]. During this stage, the infant’s brain rapidly expands to approximately 80% of its adult capacity [14], and the central nervous system (CNS) grows and expands at a rapid rate [15], which is important for the infant’s cognitive, behavioral, and emotional development. In fact, optimal stimulation, especially nutrition, during this special period is essential to ensure that infants reach their maximum neurodevelopmental potential [16,17].

Good nutrition lays the foundation for the development of cognitive, motor, and socio-emotional skills throughout childhood and adulthood, as evidenced by the fact that well-nourished children are better able to interact with their caregivers and environment [18]. Breastfeeding, which is considered to be the best way of feeding infants, could enhance cognitive growth via various potential mechanisms linked to both breast milk composition and breastfeeding experience. Positive links between extended breastfeeding duration and increased IQ and academic success were evident in cluster-randomized trials, even when adjusting for potential variables [19,20].

For infants and young children who cannot be breastfed or whose breast milk supply is insufficient for objective reasons, it is difficult for a single type of nutrient to meet the needs of brain and nerve development. Therefore, it is particularly important to supplement with multi-nutrients containing various ingredients. Formula can be a good carrier of complex nutrients so as to ensure the brain and neurocognitive development of young children from 0 to 2 years old [21]. In preadolescents, a 9-month randomized, parallel-group, double-blinded trial demonstrated that consuming foods containing added milk powder instead of snacks or meals enhanced inhibitory control and selective attention [22]. Therefore, it is crucial to study the nutritional requirements of infants and young children and to propose new early-life nutritional strategies. The focus of this study was to explore the effects of different formula components of infant formula on brain development in weanling rats.

In order to study the effects of different formula compositions of infant milk powder on the brain development of SD rats, the brain/body weight ratio of neonatal rats fed different formula milk powder were determined. The results showed the brain/body weight of rats in the 20% ordinary milk powder group compared with the basic diet group. In addition, the brain/body weight ratio of the rats in all four groups to which milk powder was added tended to increase, indicating that the supplementation of milk powder was beneficial to the growth and development of the brain of weaned rats and could increase brain weight.

5-HT, which is mostly produced by enterochromaffin cells (ECs) in the gut, plays a role in many GI functions and in sending signals from the gut to the CNS [23]. 5-HIAA is the end product of the tryptophan (TRP)-5-HT metabolic pathway and is converted from 5-HT by monoamine oxidase (MAO) [24]. 5-HT and its metabolite 5-HIAA play a crucial role in the regulation of neural activity. In this experiment, there were no significant differences in 5-HT contents among the five groups, but there was a decrease in the serum content of 5-HIAA of rats in the group supplemented with 30% ordinary milk powder. Considering that 5-HT is an intermediate product of metabolism, its content in serum is greatly interfered by environmental factors, which may not be sufficient to represent the levels of 5-HT contents in other tissues of rats. The serum levels of 5-HIAA were lower in the rats in the group with 30% ordinary milk powder, but this phenomenon did not occur in the rats with special formula milk powder. This suggests that the addition of special formula milk powder is beneficial to the intestinal flora of children, which, in turn, is beneficial to the neurological and cerebral development of children.

The tissue section stained with Hematoxylin and Eosin (H&E) is essential for making diagnoses in anatomical pathology. This staining method highlights the nucleus and cytoplasm in distinct colors, making it easier to distinguish between different cellular parts [25]. DTI, magnetic resonance imaging technique, quantitatively analyzes water molecule diffusion in three-dimensional space by measuring intensity in all directions [26]. The magnitude of FA ranges from 0 to 1, with 0 indicating that the directions are identical and 1 indicating the limiting value of the difference in diffusion direction. In MRI DTI scanning, the measurement of FA can more clearly and intuitively show the degree of damage to various fiber bundles, and the more the FA value decreases, the more serious the damage to brain white matter fibers [27]. MD embodies the microscopic movement state of all water molecules, and every possible factor that may image the movement of water molecules may affect the MD value, which is a measure of the diffusion index of the whole local tissues [28]. The decrease in the value of ADC represents the shrinkage of the extracellular space, and the movement of water molecules is obvious. A decrease in ADC represents a decrease in the extracellular space and a significant slowing of water molecule movement [29]. As can be seen from the HE staining results and NMR results, the brain tissue structure of the rats in each group was basically normal without damage, and the difference in the counts of neuronal cell degeneration and necrosis in the brain tissue of the rats in each group was not statistically significant. Therefore, the nutrients and active ingredients added to the special formula milk powder have no damaging effect on the rat brain.

Lipids play vital roles in the human body as they are found everywhere and serve as crucial components of cell membranes and hormones [30]. In recent times, lipids have expanded their function beyond only membrane structure and energy storage to also include intercellular signaling, growth regulation, cellular differentiation, commitment, and the maintenance of cellular homeostasis [31]. In the human body, adipocytes and the brain are the most lipid-rich organs [32]. Analysis of the KEGG metabolic pathway shows that lipids are involved in multiple metabolic processes in the brain. Lipidomics is an analytical method that allows for the identification and quantification of lipids throughout an organism [33]. The analysis of lipidomic data from the brain allows us to better understand neurological functions and diseases and to propose new therapeutic approaches based on metabolic mechanisms and responses [34].

In the present study, we investigated the lipid profiles of brain tissues of SD rats supplemented with different formula compositions of infant milk powder by a non-targeted lipidomic approach. As shown in Figure 5, through the Venn plot, PCA analysis, and PLS-DA analysis, the lipid metabolite species of the brain tissues of the five groups of rats showed a high degree of similarity, and there were more common lipid species. The differences in lipid metabolite species between the group with 20% special formula milk powder and the other groups were large, and MePC (22:6/14:3) was a lipid metabolite specific to the brain tissues of the rats in the group with 20% special formula milk powder. Therefore, the effects of different dosages and formula compositions of infant milk powder on the lipid metabolite species of rat brain tissue in each group were small.

Relatively high expressions of GP metabolites such as DLCL, MePC, and PI in brain tissues of SD rats were in the groups supplemented with 20% and 30% special formula milk powder. GP is a molecule that is amphiphilic and actively involved in the function of ion channels, as well as transporters, receptors, and the regulation of neuronal membrane function and proliferation. Cerebral glycerophospholipids have been reported to be reduced whenever AD pathology is present [35,36]. DLCL is a metabolite of CL and one of the least abundant phospholipids [37], for which there is a lack of relevant research. However, an animal experiment on a mouse model of AD showed a reduction in the total amount of CL associated with mitochondrial synaptic dysfunction and oxidative stress in the brains of AD mice, indicating that CL has an impact on the pathogenesis of the disease [38], and a potential protective role of CL in the AD inflammatory reaction has been suggested [39]. Multiple studies have found reductions in phosphatidylinositols (PIs) in AD brains compared with healthy controls [40,41]. Choline, abundant in the special formula milk powder, serves as a building block for membrane phospholipids like PCs. In animal models, high intake of choline during the early postnatal period enhances cognitive abilities in adulthood, guards against memory loss with age, and shields the brain from the pathological alterations linked to AD [42]. Moreover, the addition of phospholipid in the special formula may also play a role, so that special formula can increase the beneficial lipids in the brains of rats. However, the relative concentration of LPE was lower in the 20% special milk powder group than in the basic diet group. Analysis of brain samples from AD patients showed that LPE expression was lower in the brains of patients with better Cerad scores [43]. Meanwhile, Llano et al. stated that a high level of LPE leads to a twofold increase in the speed of progression from mild cognitive impairment (MCI) to AD [44]. Thus, the beneficial effects of special formulas on the rat brain and cognition may be through modulation of the expression of different types of GP in the brain.

Meanwhile, GM1 expression was relatively high in the brain tissue of both groups of rats. GM1, a significant glycosphingolipid (GSL) located on the surface of neural cells in the brain, shields neural tissues from harmful compounds and sustains neuronal functions [36]. Moreover, GM1 is essential for cognition and memory processes, as evidenced by in vivo experiments demonstrating that external GM1 triggers the release of neurotransmitters from synaptosomes in the cortex of mice [45]. Therefore, increasing the concentration of GM1 in the central nervous system is expected to offer benefits for a wide range of neurological conditions such as AD, PD, Huntington’s Disease (HD), depression and anxiety, epilepsy, and seizures [36]. The expression of GM1 in brains in the 30% special milk powder group was higher than the basic diet group, which indicated that special formula could increase the expression of GM1 in the brains of rats. The addition of ganglioside in the special formula may play a major role. Studies showed that gangliosides from the diet could be taken in by the small intestine and notably raised the overall ganglioside levels in the brain [46]. Findings from animal experiments indicated that adding gangliosides to the diet of female rats during gestation can significantly enhance the amounts of brain gangliosides in the offspring shortly after delivery [47]. SM is altered in neurodegeneration [48], and the accumulation of SM in the brain activates γ-secretase activity. This leads to elevated production of both intracellular and secreted Aβ, contributing to neurodegeneration in sporadic AD [49]. Previous studies indicated that SM levels were elevated in the brains of individuals with AD and correlated with the clinical symptoms [43]. The expression of SM of the brain in the 30% ordinary milk powder group was higher than in the basic diet group, while the differences in the groups with added special formula milk powder were not significant, which indicated that special formula could reduce the adverse effects on cognitive function.

Previous research highlighted increased levels of DG and TG lipid species in both AD and mild AD PM cohorts when compared with controls [35]. In this experiment, the expression of TG and DG of GL indicated almost no differences between the five groups, but the relative concentrations of MGDG in the 20% ordinary milk powder group and 30% ordinary milk powder group were higher than those in the basic diet. MGDGs are a group of neutral GL present in minimal quantities within the mammalian nervous system. Due to the peak levels of their biosynthesis and the highest levels occurring during the period of maximum myelin production in postnatal development, they have been suggested as indicators of myelin and myelination [50]. A study demonstrated that MGDG levels in the brain increased substantially as Alzheimer’s disease advanced by assessing their levels in the brain of deceased individuals [51]. The result of MGDG indicates that adding high-dose special milk powder may decrease the expression level of MGDG.

Previous research showed that excessive levels of saturated fatty acids are toxic to cells [52]. The results of the present study showed that adding special formula milk powder could elevate the saturation of brain lipids and the proportion of lipids with low unsaturation, which was beneficial to the development of the brain.

Overall, the beneficial effects of special formulations on the rat brain and cognitive abilities may be achieved by modulating the expression of different types of lipids in the brain. This may be due to the special formula referring to milk powder with the appropriate amounts of DHA, ARA, galactose oligosaccharide, 1,3-dioleic acid-2-palmitate triglyceride, lutein, nucleotide, lactoferrin, casein phosphopeptide. Experiments on the basis of cells and animals support the use of DHA to enhance brain function, specifically for promoting the growth and differentiation of neuronal cells, as well as improving neuronal signaling pathways [53]. The proportions of arachidonic acid (ARA) have an important involvement in early development, including immune cells, astrocyte development, and neural connectivity [54]. Oligosaccharides and polyunsaturated fatty acids (such as DHA and ARA) have been directly linked to the development of infants’ gut microbiota–gut–brain axis and are noted to promote infant brain development and gut microbiota formation [55,56], and dietary nucleotides in milk may affect neurodevelopment and maturation in early life by regulating the gut microbiota composition–gut–brain axis [56]. Lutein accounts for more than half of the total carotenoid concentration in the infant brain, which is twice that of adults, indicating that large amounts of lutein are required during the neural development of newborns [57]. Therefore, adding these nutrients to the formula may promote the brain development of infants.

According to the WB results, synapse-associated protein Postsynaptic density protein-95 (PSD-95) expression did not differ significantly among the five groups. PSD-95 mainly regulates synapse maturation by interacting, and it has been confirmed that the decline in PSD-95 level is related to cognitive and learning deficits observed in autism by obvious evidence [58]. Therefore, ordinary and special formulas have no negative effect on the expression of synaptic proteins in the brain. The expression of Myelin PLP (PLP), a myelin-related indicator, did not differ significantly among the five groups of rats, whereas the expression of MBP was higher in the brain tissues of the rats in the group supplemented with 20% of ordinary milk powder, compared with lower expression in the brain tissues of rats in the basic diet group, the 30% ordinary milk powder group, and the 30% special milk powder group. MBP is associated with the formation of myelin in the central nervous system and has long been implicated as a factor in the pathogenesis of the autoimmune neurodegenerative disease multiple sclerosis (MS) [59,60]. Therefore, the special ingredients in the formula do not impair brain function and are beneficial to brain function.

One concern about the findings was that the simulation of human infants may not be accurate due to the excessive difficulty of artificially feeding unweaned rats, which were studied in this study using weaned rats through the addition of different concentrations of different types of powdered milk to the chow. In spite of this limitation, it does not affect the changes in lipidomics of rat brains from the special formula milk powder in the results. These findings provide the following insights for future research: we will use unweaned animals for more accurate modeling of human infant growth and development.

## 5. Conclusions

In this study, we found that different formula components in infant milk powder could affect the changes in brain/body weights, serum 5-HIAA, brain lipidomics, and MBP protein expression levels in rats. Meanwhile, the special formula milk powder may have beneficial effects on rat brains by regulating brain lipid expression.

## Figures and Tables

**Figure 1 nutrients-16-02188-f001:**
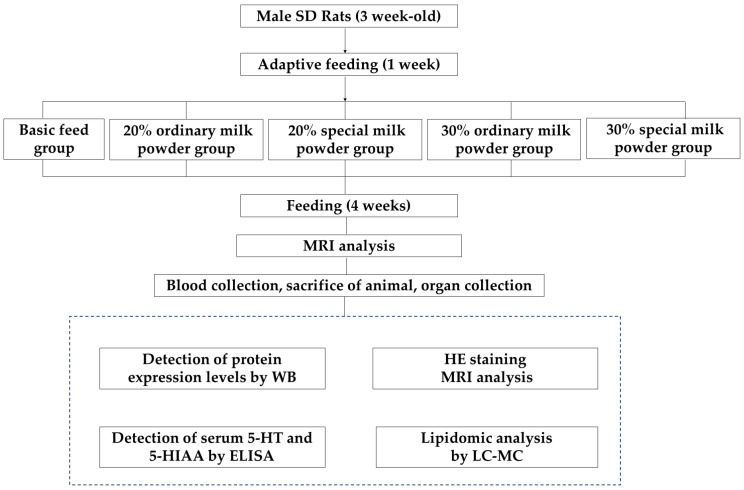
The flow chart of experimental design.

**Figure 2 nutrients-16-02188-f002:**
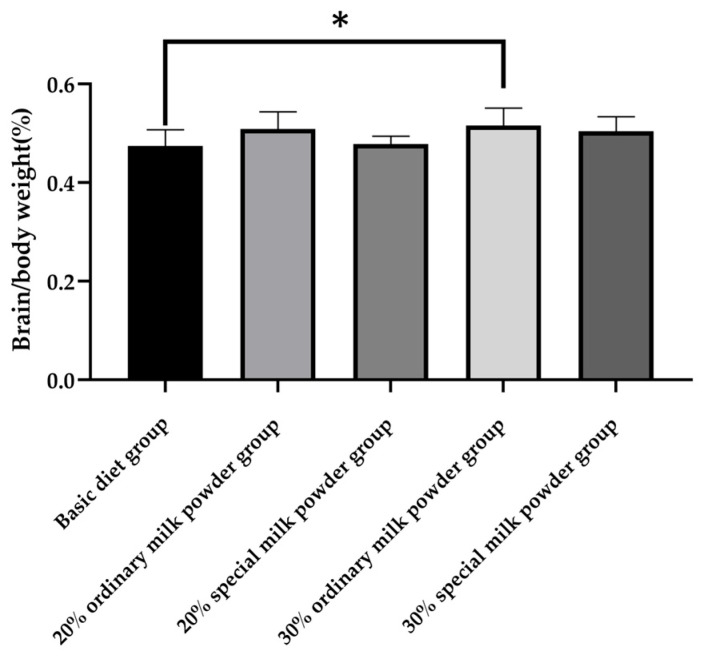
The brain/body weights of rats fed ordinary milk powder and special formula milk powder (n = 10); * *p* < 0.05.

**Figure 3 nutrients-16-02188-f003:**
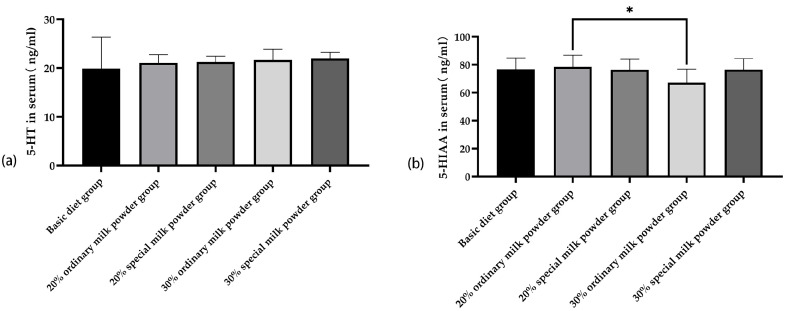
The serum 5-HT and 5-HIAA of rats fed ordinary milk powder and special formula milk powder (n = 10). (**a**) the serum 5-HT; (**b**) the serum 5-HIAA; * *p* < 0.05.

**Figure 4 nutrients-16-02188-f004:**
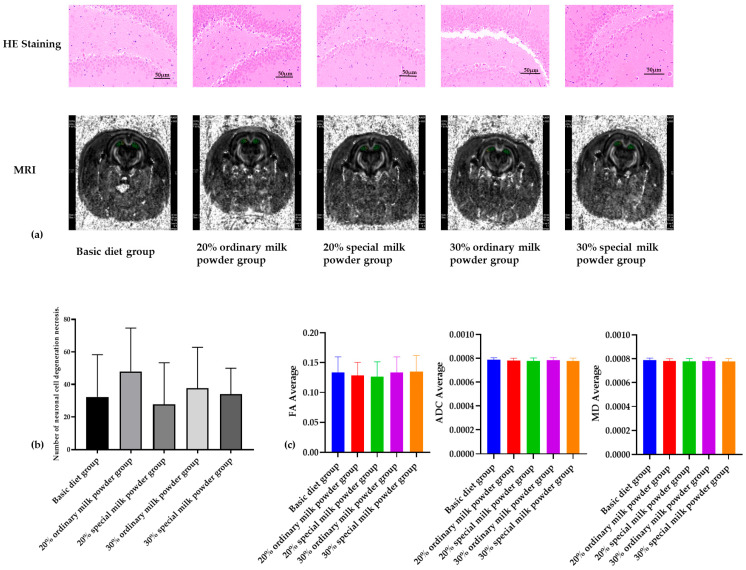
The structure of brain of rats fed ordinary milk powder and special formula milk powder (n = 3). (**a**) the HE staining and MRI image of brain; (**b**) counts of neuronal cell degeneration and necrosis in brain tissues; (**c**) structural analysis of brain.

**Figure 5 nutrients-16-02188-f005:**
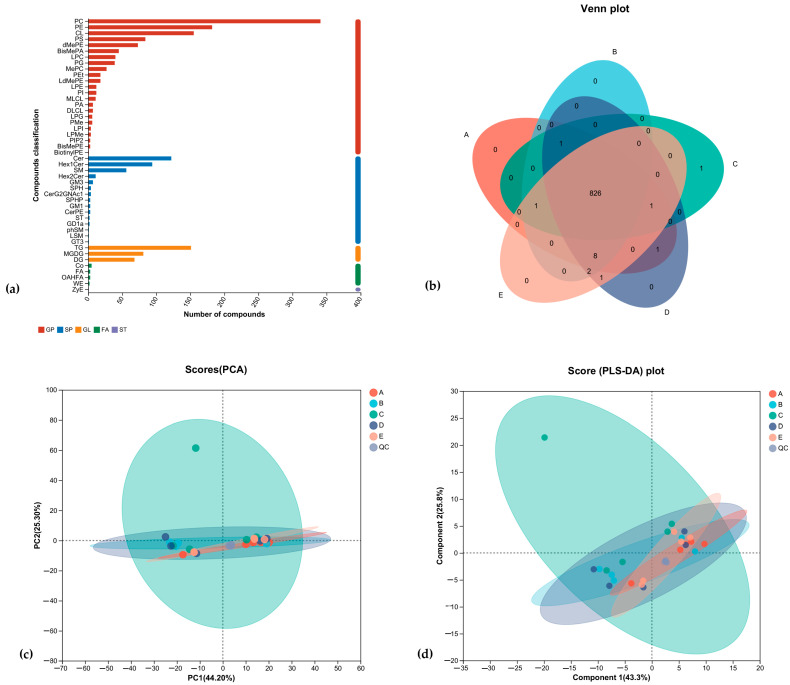
The comparative analysis of lipid metabolites of rats fed ordinary milk powder and special formula milk powder (n = 5). A, basic diet group; B, 20% ordinary milk powder group; C, 30% ordinary milk powder group; D, 20% special milk powder group; E, 30% special milk powder group; QC, quality control group. (**a**) lipid classification statistics of brain tissue; (**b**) Venn diagram of brain tissue; (**c**) lipid PCA analysis; (**d**) lipid PLS-DA analysis.

**Figure 6 nutrients-16-02188-f006:**
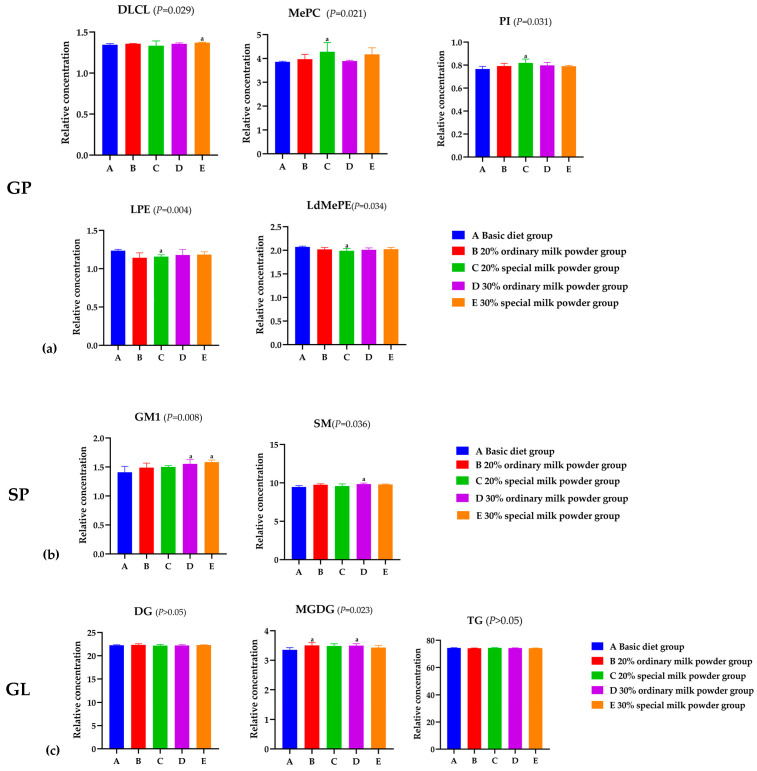
Analysis of differential lipid metabolites of rats fed ordinary milk powder and special formula milk powder. (n = 5). (**a**) the relative concentration of GP in different groups; (**b**) the relative concentration of GL in different groups; (**c**) the relative concentration of SP in different groups. a: *p* < 0.05, compared to the basic diet group.

**Figure 7 nutrients-16-02188-f007:**
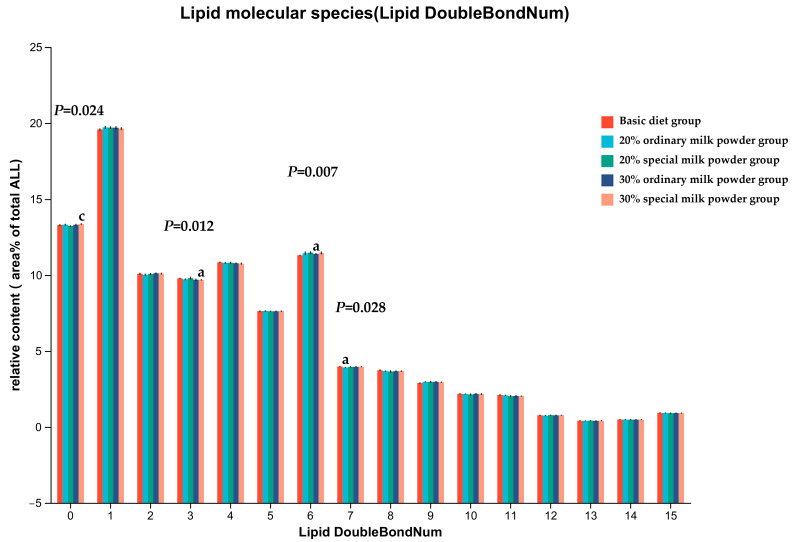
Unsaturation degree of total FA in brain of rats fed ordinary milk powder and special formula milk powder. a: *p* < 0.05, compared to the basic diet group; c: *p* < 0.05, compared to 20% special milk powder group; Mean ± SD; n = 5.

**Figure 8 nutrients-16-02188-f008:**
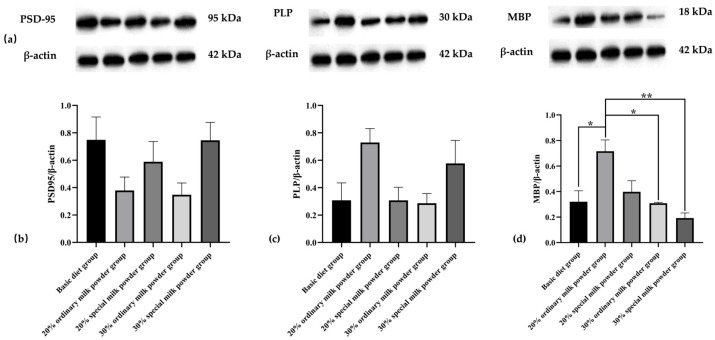
The brain protein expression of rats fed ordinary milk powder and special formula milk powder (n = 3). (**a**) Western-blotting bands in SD rat brain tissue; (**b**) PSD-95/β-actin relative expression; (**c**) PLP/β-actin relative expression; (**d**) MBP/β-actin relative expression; * *p* < 0.05, ** *p* < 0.01.

**Table 1 nutrients-16-02188-t001:** Ordinary milk powder and special milk powder.

Special Ingredients	Special Milk Powder	Ordinary Milk Powder
α-linolenic acid (mg)	260	-
Selenium (μg)	12	-
choline (mg)	120	42.8
Manganese (μg)	37	-
inositol (mg)	40	25.3
taurine (mg)	38	19
L-carnitine (mg)	11	7.8
DHA (mg)	50	-
Eicosatetraenoic acid (ARA)	85	-
Galactooligosaccharide (mg)	3	-
1,3-dioleic-2-palmitate triglyceride (g)	4	-
Lutein (g)	210	-
nucleotide (μg)	30	-
lactoferrin (mg)	45	-
casein phosphopeptides (mg)	40	-
Animal Bifidobacterium bb-12 (CFU)	109	-
Ganglioside (mg)	67.1	-
total phospholipid (mg)	348	-

**Table 2 nutrients-16-02188-t002:** Basic diet and experimental diet.

	Basic Diet	Ordinary Milk Powder Diet (20%)	Special Milk Powder Diet (20%)	Ordinary Milk Powder Diet (30%)	Special Milk Powder Diet (30%)
kcal/g	3.99	3.92	3.93	3.89	3.90
Protein	17.3	17.3	17.3	17.3	17.3
Fat	18.8	20.4	21.1	19.5	19.8
Carbohydrate	63.9	62.3	61.6	63.2	62.9
Casein (g)	20	15.9	16.5	13.8	14.7
L-Cystine (g)	0.3	0.3	0.3	0.3	0.3
Corn Starch (g)	39.7	27.7	27.2	21.8	21.0
Maltodextrin (g)	13.2	13.2	13.2	13.2	13.2
Sucrose (g)	10	10	10	10	10
Cellulose (g)	5	5	5	5	5
Soybean oil (g)	7	3.1	3	1.15	1
Hydrocholine Bitartrate (g)	0.25	0.25	0.25	0.25	0.25
Mineral Mix (g)	3.5	3.5	3.5	3.5	3.5
Vitamin Mix (g)	1	1	1	1	1

**Table 3 nutrients-16-02188-t003:** Statistical table of KEGG pathway in rat brain tissue.

Pathway ID	Pathway Description	First Category	Second Category	Number of Lipid Metabolites
map01100	Metabolic pathways	Metabolism	Global and overview maps	90
map00564	Glycerophospholipid metabolism	Metabolism	Lipid metabolism	73
map05231	Choline metabolism in cancer	Human Diseases	Cancer: overview	69
map04723	Retrograde endocannabinoid signaling	Organismal Systems	Nervous system	54
map04931	Insulin resistance	Human Diseases	Endocrine and metabolic disease	39
map04975	Fat digestion and absorption	Organismal Systems	Digestive system	32
map00561	Glycerolipid metabolism	Metabolism	Lipid metabolism	25
map04071	Sphingolipid signaling pathway	Environmental Information Processing	Signal transduction	19
map00600	Sphingolipid metabolism	Metabolism	Lipid metabolism	12
map05200	Pathways in cancer	Human Diseases	Cancer: overview	8

## Data Availability

The raw data supporting the conclusions of this article will be made available by the authors upon request.

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
