# Peer review of "Diet Supplemented with Special Formula Milk Powder Promotes the Growth of the Brain in Rats"

_nutrients, 2024, doi:10.3390/nu16142188_

Round 1
Reviewer 1 Report
Comments and Suggestions for Authors
Mu and colleagues investigated the impact of different formula to the growth of rat brain and its contents to find that fortification of different contents and degrees resulted in altered growth of brain and its composition. The study design is comprehensive to depict the change in brain conformation, quantity and quality.
Major points:
It was difficult to follow the text because of excessive use of non-standard abbreviations. Some including IF and HM were used without proper definition, Most of these do not worth abbreviating. How can the authors use an acronym of FA for two different terms?
It is inappropriate to use the term development as this study only investigated the growth, but not function. The term in the title, abstract and text should be replaced by growth throughout.
I wonder why the authors did not perform neurological functional assessment before sacrificing the rats.
Introduction:
The background section is easy to follow but failed to highlight why the five study groups representing different formula types were chosen. The background should be deeply, directly connected with the study question.
Materials and Methods:
The authors need to clarify how many litters were used. Because only ten each rat pups belong to a group, genetic backgrounds and maternal milk expression before weaning are crucially important. Imagine if eight of ten rats in one group came from a same mother, the size of which is much bigger than others and expressed relatively more milk. The information of the litter should be incorporated within the analysis.
Age at the commencement of study is widely ranged between 21 and 27 days, which represents months in human infants.
The difference and concept of the five groups were difficult to understand, which should not be expressed by words like “special”. Please be more specific to highlight why the specific formula was chosen. Perhaps an additional table highlighting the difference in the content will help.
Figure 1 was difficult to follow especially after sacrificing the rat. Flow chart is useful to show the time line or order of procedures, but not in the current case.
Because the brain weight is the primary outcome of the study, the authors need to describe in detail exactly when and how the brain was harvested. The length of the brain stem part, removal of blood and procedures of washing/perfusing significantly affect the outcome. Also the authors should describe how the brain was divided and which part was used for each analysis. Was MRI performed in a frozen slice or after fixation? Which section?
Because numerous comparisons have been performed in five experimental groups, the threshold of p < 0.05 is generally too optimistic to identify significant differences. Especially for ANOVA, appropriate post hoc tests should be used to adjust for multiple comparisons.
Results:
Once again, p-values are generally close to the threshold.
This study does not apply hypoxia or vessel occlusion. Then what do brain lesions and neuronal degeneration mean?
Discussion:
The Discussion section is well written but several sections are excessively speculative. Please minimize jumps between findings and interpretation.
The Conclusion section should be written objectively. Please avoid using terms like “beneficial lipids” and “pathological lipids”.
Minor points:
There are numerous typo errors and undeleted instructions, such in line 27, page “it is it is”, line 32, page 11 “authors should discuss the…”.
Approximately 10% of necessary spacing is lacking in the text e.g. between words and parenthesis, period and next sentence and comma and words.
Comments on the Quality of English Language
Many typo erros and immature use of spacing.
Author Response
Comments 1: It was difficult to follow the text because of excessive use of non-standard abbreviations. Some including IF and HM were used without proper definition, Most of these do not worth abbreviating. How can the authors use an acronym of FA for two different terms? |
Response 1: Thank you for pointing this out. We agree with this comment. Therefore, we have checked and removed most of the unnecessary abbreviations.
|
Comments 2: It is inappropriate to use the term development as this study only investigated the growth, but not function. The term in the title, abstract and text should be replaced by growth throughout. |
Response 2: Agree. We have, accordingly, replaced the word "development" with the word "growth" throughout the text.
Comments 3: I wonder why the authors did not perform neurological functional assessment before sacrificing the rats. Response 3: Thanks for pointing out this omission, which we did make in our assessment of neurological function. Neurobehavioral assessments are usually used in models of neurological disease and are somewhat damaging to rats, while this study assessed the functional effects of specific nutrients in the formula on the brain, and therefore focused on studies that more accurately reflect brain structure and function, such as lipidomics. And we will take care to add the neurological functional assessment to our future research.
Comments 4: The background section is easy to follow but failed to highlight why the five study groups representing different formula types were chosen. The background should be deeply, directly connected with the study question. Response 4: Thank you for pointing this out. We have revised the introduction, and the reason for the five groups is that the experiment was designed to investigate whether the dose of milk powder had an effect on brain growth, but the results showed that the effect of the dose was not significant, so we did not highlight and discuss it.
Comments 5: The authors need to clarify how many litters were used. Because only ten each rat pups belong to a group, genetic backgrounds and maternal milk expression before weaning are crucially important. Imagine if eight of ten rats in one group came from a same mother, the size of which is much bigger than others and expressed relatively more milk. The information of the litter should be incorporated within the analysis. Response 5: Thank you for pointing this out. Unfortunately, however, we were unable to take this information into account in our analysis due to the fact that we did not retain information about the genetic background of these rats and their mothers, only that the 50 rats came from at least 6 litters. However, we grouped the rats randomly by weight, and since it is the weight of the rats that is most likely to be affected by differences in mother's milk, we believe that this grouping will offset the differences in coming from different litters. However, we will pay attention to this issue in future studies.
Comments 6: Age at the commencement of study is widely ranged between 21 and 27 days, which represents months in human infants. Response 6: Thank you for pointing this out. We have added these to the methods section. The weekly age of newly weaned rats was 3-4 weeks, corresponding to the regular human weaning age of about 6 months, and the feeding period was 4 weeks, corresponding to the human weaning to prepubertal age.
Comments 7: The difference and concept of the five groups were difficult to understand, which should not be expressed by words like “special”. Please be more specific to highlight why the specific formula was chosen. Perhaps an additional table highlighting the difference in the content will help. Response 7: Thank you for pointing this out. We have, accordingly, added a table to illustrate the difference between regular and special formulas
Comments 8: Figure 1 was difficult to follow especially after sacrificing the rat. Flow chart is useful to show the time line or order of procedures, but not in the current case. Response 8: Thank you for pointing this out. We have modified Figure 1 based on the timeline.
Comments 9: Because the brain weight is the primary outcome of the study, the authors need to describe in detail exactly when and how the brain was harvested. The length of the brain stem part, removal of blood and procedures of washing/perfusing significantly affect the outcome. Also the authors should describe how the brain was divided and which part was used for each analysis. Was MRI performed in a frozen slice or after fixation? Which section? Response 9: Thank you for pointing this out. We have added methods for taking brain tissue to the methods section: The entire brain was rinsed repeatedly with 0.9% saline, then dried on filter paper, and weighed. (line 107). Unfortunately, however, we did not measure the length of the brainstem, so we are unable to provide this part of the data, and we will pay attention to this in the future studies. In addition, we performed MRI in rats under anesthesia (line 96).
Comments 10: Because numerous comparisons have been performed in five experimental groups, the threshold of p < 0.05 is generally too optimistic to identify significant differences. Especially for ANOVA, appropriate post hoc tests should be used to adjust for multiple comparisons. Response 10: Thank you for pointing this out. We have looked for information and reanalyzed the data through a more scientific approach: One-way ANOVA was used for data that conformed to normality and had chi-square variance, then Tukey’s multiple comparisons test method for post hoc test, and non-parametric tests (Kruskal-Wallis test) were used for data that did not conform to normality, and Welch method and Dunnett’s T3 method were used for data that conformed to normality but didn’t have chi-square variance, where P<0.05 indicated a statistically significant level of significance of the differences.
Comments 11: T This study does not apply hypoxia or vessel occlusion. Then what do brain lesions and neuronal degeneration mean? Response 11: Thank you for pointing this out. When designing the experiment, we wanted to know whether a single nutrient, such as a basis diet, had a negative effect on brain structure as the rats grew, and whether the addition of a specially formulated formula would reverse these effects, so MRI and HE staining were added to examine the rat brains for significant lesions and neuronal degeneration.
Comments 12: The Conclusion section should be written objectively. Please avoid using terms like “beneficial lipids” and “pathological lipids”. Response 12: Thank you for pointing this out. We have modified the conclusion section: the special formula milk powder may have beneficial effects on rat brain by regulating brain lipid expression.
Comments 13: There are numerous typo errors and undeleted instructions, such in line 27, page “it is it is”, line 32, page 11 “authors should discuss the…”. Approximately 10% of necessary spacing is lacking in the text e.g. between words and parenthesis, period and next sentence and comma and words. Response 13: Thank you for pointing this out. We have, accordingly, fixed the content and formatting errors in the article. |

Reviewer 2 Report
Comments and Suggestions for Authors
This is an interesting paper that describes brain/body weight, fatty acid levels in brain and brain structure in response to different diets. The major problem is the diets are not defined. Line 43 mentions soy. Was soy protein used in the study? If so, was there a casein control group? And soy bioactives should be discussed. Soy is common but not “typically utilized as foundational element”. What was the vivarium chow prior to changing diets? What were the mice fed for 4 weeks before randomization to groups? Need a table listing all ingredients in all diets. What is ordinary milk powder? What is adaptive feeding? How does transitioning rats to new diets at 3 or 4 weeks old mimic infant formula feeding and its effects on brain development?
Why would normal rats have brain lesions or neuronal degeneration?
The writing on the graphs is small. Would be helpful to show individual data points versus bars. The writing on Figure 7 is particularly small and blurry.
Since the objective is to study the effect of diet on brain development, why are there no measures of brain development such as behavioral/cognitive testing?
Needs minor English editing, for example, we don’t “execute” mice.
Comments on the Quality of English LanguageSeveral instances of strange word use throughout the manuscript, for example, "executing" mice.
Author Response
Comments 1: This is an interesting paper that describes brain/body weight, fatty acid levels in brain and brain structure in response to different diets. The major problem is the diets are not defined. Line 43 mentions soy. Was soy protein used in the study? If so, was there a casein control group? And soy bioactives should be discussed. Soy is common but not “typically utilized as foundational element”. What was the vivarium chow prior to changing diets? What were the mice fed for 4 weeks before randomization to groups? Need a table listing all ingredients in all diets. What is ordinary milk powder? What is adaptive feeding? How does transitioning rats to new diets at 3 or 4 weeks old mimic infant formula feeding and its effects on brain development? |
Response 1: Thank you for pointing this out. We agree with this comment. Therefore, we have added a table to illustrate the difference between regular and special formulas. It is mentioned in the introduction (Line 44) that “Soy milk or milk is typically utilized as the foundational element of formula”, and in our research we use milk, not soy milk; Rats are breastfed for 1-3 weeks, corresponding to the human lactation period (0-6 months). They were then acclimatized with the basic diet for another week. Then, they were grouped according to body weight and fed for 4 weeks, corresponding to the postweaning to prepubertal period in humans.
|
Comments 2: Why would normal rats have brain lesions or neuronal degeneration? |
Response 2: Thank you for pointing this out. When designing the experiment, we wanted to know whether a single nutrient, such as a basic diet, had a negative effect on brain structure as the rats grew, and whether the addition of a specially formulated formula would reverse these effects, so MRI and HE staining were added to examine the rat brains for significant lesions and neuronal degeneration.
Comments 3: The writing on the graphs is small. Would be helpful to show individual data points versus bars. The writing on Figure 7 is particularly small and blurry. Response 3: Thank you for pointing this out. We have, accordingly, fixed font size and clarity for Figure 7.
Comments 4: Since the objective is to study the effect of diet on brain development, why are there no measures of brain development such as behavioral/cognitive testing? Response 4: Thanks for pointing out this omission, which we did make in our assessment of neurological function. Neurobehavioral assessments are usually used in models of neurological disease and are somewhat damaging to rats, while this study assessed the functional effects of specific nutrients in the formula on the brain, and therefore focused on studies that more accurately reflect brain structure and function, such as lipidomics. However, we will add the behavioral tests in the future studies.
Comments 5: Needs minor English editing, for example, we don’t “execute” mice. Response 5: Thank you for pointing this out. We have, accordingly, edited the language of the article, and we will pay attention to this in the future. |

Round 2
Reviewer 1 Report
Comments and Suggestions for Authors
The authors have worked hard to improve the manuscript.
Author Response
We sincerly appreciate the valuable comments from reviewer. Hearfelt thanks for your hardwaok!
Reviewer 2 Report
Comments and Suggestions for Authors
Two major problems remain:
(1) The authors have not explained how the dosing strategy mimics human infant exposure or why a 7 day adaptive period is required. Rats are considered infants up to 21 days old. These mice were weaned at 22-27 days, underwent 7 days of adaptive feeding and then 4 weeks of formula treatments, which equates with formula feeding from about 29/34 days of age to 57/62 days of age. The human equivalent of childhood in rats is 22-35 days and adolescence is 55-70 days. How does this mimic human exposure to infant formula?
(2) Table 1 only lists special ingredients. What are the base ingredients (amount and caloric contribution) of the test diets and the adaptive diet?
Other:
Line 44 still mentions soy as foundation element of infant formula. In the U.S., soy formula use is estimated at about 12%, i.e., not the major protein used in infant formula. The response states soy milk is not used. Why is this mentioned? Does the adaptive diet contain soy? If so, how is that confounding the results?
An explanation is still not provided as why brain lesions would be expected in normal rats. What ingredient in the base diet is expected to cause brain lesions?
Author Response
Comments 1: The authors have not explained how the dosing strategy mimics human infant exposure or why a 7 days adaptive period is required. Rats are considered infants up to 21 days old. These mice were weaned at 22-27 days, underwent 7 days of adaptive feeding and then 4 weeks of formula treatments, which equates with formula feeding from about 29/34 days of age to 57/62 days of age. The human equivalent of childhood in rats is 22-35 days and adolescence is 55-70 days. How does this mimic human exposure to infant formula? |
Response 1: Thank you for pointing this out. We agree with this comment. First, the reason for the 7-day acclimatization period was the concern that the rats would not be able to adapt to the new environment leading to stress, which would affect the experimental results. Therefore, we used adaptive diet to allow them to adapt to the environment of the animal experiment center before intervention. Secondly, milk powder intake was calculated based on infant milk powder intake from 0 to 6 months of age, corresponding to the body weight of the rats. Finally, it was indeed a flaw in our experimental design that, as it was too difficult to achieve an intervention of milk powder in unweaned pups, so we had to wait until the rats were weaned. Besides, we also reviewed some formula-related studies using weaned rats in our experimental design[1-3]( References are at the end of the reply). We have also added a description of this flaw in the discussion section (line 522). In future experiments, we will actively explore artificially feeding unweaned rats, thus refining the research. |
Comments 2: Table 1 only lists special ingredients. What are the base ingredients (amount and caloric contribution) of the test diets and the adaptive diet? |
Response 2: Thank you for pointing this out. We have added a new table to illustrate the caloric contribution and formulation of the five diets. Adaptive feeding diets and basic diets are based on the purified feed standards for rodent laboratory animals promulgated by the American institute of Nutition (AIN) and are widely used in different types of research. Besides, we made the energy density of the five diets the same by adjusting the other conventional formulations of the diets.
Comments 3: Line 44 still mentions soy as foundation element of infant formula. In the U.S., soy formula use is estimated at about 12%, i.e., not the major protein used in infant formula. The response states soy milk is not used. Why is this mentioned? Does the adaptive diet contain soy? If so, how is that confounding the results? Response 3: Thank you for pointing this out. There is no soy in the adaptive diet. We have, accordingly, removed the part about soy milk from the introduction.
Comments 4: An explanation is still not provided as why brain lesions would be expected in normal rats. What ingredient in the base diet is expected to cause brain lesions? Response 4: Thanks for pointing out this out. Since there was no obvious structural damage in the MRI examination of the rats and the difference in the counts of neuronal cell degeneration and necrosis in the brain tissue of the rats in each group was not statistically significant, we concluded that slight neuronal degeneration is a normal phenomenon during the growth process of the rats, which may be caused by environmental (e.g., noise) and psychogenic (e.g., stress) factors.
|
- Calvez, J.; Blais, A.; Deglaire, A.; Gaudichon, C.; Blachier, F.; Davila, A.M. Minimal processed infant formula vs. conventional shows comparable protein quality and increased postprandial plasma amino acid kinetics in rats. The British journal of nutrition 2024, 131, 1115-1124, doi:10.1017/s0007114523002696.
- Liu, Z.; Subbaraj, A.; Fraser, K.; Jia, H.; Chen, W.; Day, L.; Roy, N.C.; Young, W. Human milk and infant formula differentially alters the microbiota composition and functional gene relative abundance in the small and large intestines in weanling rats. European Journal of Nutrition 2020, 59, 2131-2143, doi:10.1007/s00394-019-02062-w.
- Liu, Z.; Roy, N.C.; Guo, Y.; Jia, H.; Ryan, L.; Samuelsson, L.; Thomas, A.; Plowman, J.; Clerens, S.; Day, L., et al. Human Breast Milk and Infant Formulas Differentially Modify the Intestinal Microbiota in Human Infants and Host Physiology in Rats123. The Journal of Nutrition 2016, 146, 191-199, doi:https://doi.org/10.3945/jn.115.223552.
